# Egg Parasitoids Survey of *Spodoptera frugiperda* (Smith) (Lepidoptera: Noctuidae) in Maize and Sorghum in Central Mexico

**DOI:** 10.3390/insects11030157

**Published:** 2020-03-01

**Authors:** Jannet Jaraleño-Teniente, J. Refugio Lomeli-Flores, Esteban Rodríguez-Leyva, Rafael Bujanos-Muñiz, Susana E. Rodríguez-Rodríguez

**Affiliations:** 1Posgrado en Fitosanidad-Entomología y Acarología, Colegio de Postgraduados, Estado de México, 56230 Texcoco, Mexico; tenientejanet@gmail.com (J.J.-T.); esteban@colpos.mx (E.R.-L.); serr_biol@hotmail.com (S.E.R.-R.); 2Instituto Nacional de Investigaciones Forestales, Agrícolas y Pecuarias (INIFAP), Campo Experimental Bajío, Guanajuato, 38010 Celaya, Mexico; bujanos@live.com.mx

**Keywords:** *Trichogramma atopovirilia*, *Trichogramma pretiosum*, fall armyworm, natural enemies

## Abstract

*Spodoptera frugiperda* (Smith) is the main maize pest in America and was recently detected as an invasive pest in some countries in Asia and Africa. Among its natural enemies presented in Mexico, *Trichogramma pretiosum* Riley is the only egg parasitoid used in Integrated Pest Management (IPM) programs regardless of its effectiveness. A search for natural enemies of *S. frugiperda* was then carried out to determine whether this parasitoid has been established, and to detect native egg parasitoids or predators associated with this pest. The sentinel technique (egg masses) was used, and then placed in maize and sorghum fields in the state of Guanajuato, Mexico. *Trichogramma atopovirilia* Oatman and Platner, an egg parasitoid, and *Chelonus insularis* Cresson egg-larva parasitoid were recovered from field surveys. Among the natural enemies that preyed on eggs of *S. frugiperda*, we found mites of the genus *Balaustium*, and Dermaptera of the genus *Doru*, both species in great abundance. Laboratory tests were performed to compare the potential parasitism of *T. atopovirilia* against *T. pretiosum*. *T. atopovirilia* obtained 70.14% parasitism while *T. pretiosum*, 29.23%. In field cages, three doses of the parasitoids were tested. Total parasitism did not exceed 8% in any of the two species, but *T. atopovirilia* parasitized a greater number of hosts using two and three parasitoids per pest egg. Then, the use of *Trichogramma* species needs to be reevaluated in biological control programs against *S. frugiperda*.

## 1. Introduction

*Spodoptera frugiperda* (Smith) (Lepidoptera: Noctuidae), one of the most harmful pests in the world, limited the yield of maize, sorghum, and other crops [1]. This species is widely distributed among the Americas and has recently been reported in 43 countries throughout Africa and some in Asia, such as India, China, and Thailand [2,3,4,5]. The larva frequently attacks the early and late whorl stage feeding on the growing point, as well as leaves; if the feeding occurs in the early whorl stage it can kill the plant, but if it is in the foliage, it can occasionally recover [6,7]. *S. frugiperda* is the main maize pest in Mexico and can reduce the production of this grain by 45% [8]; and Argentina reports yield losses of up to 72% [9]. Although it has been recently introduced in Africa (2016), it has also caused losses between 20% and 50% of cultivated maize in this area [10].

The high infestation rates of *S. frugiperda*, and the economic losses that it causes, are the justification for the use of organ-synthetic pyrethroid and organophosphate insecticides as the most frequent control tactic. This action increases the selection of resistant populations, in addition to other undesirable consequences. There are reports of resistance cases of *S. frugiperda* to organochlorine compounds, organophosphates, carbamates, and pyrethroids for at least two decades in the US [11,12]. This situation was also recorded in Mexico with the same chemical groups [13], and with pyrethroids in Brazil [14].

Because *S. frugiperda* attacks crops such as maize, sorghum, cotton, and rice, and considering the total area dedicated to these crops in the world [15], it is necessary to improve the management proposal. The inclusion of biological control, as one of the tactics used for the integrated management programs of this pest, is desirable.

Mexico is the center of origin and domestication of maize [16], with a great diversity of ecosystems [17], this is why there is a high possibility of finding native natural enemies associated with *S. frugiperda* [18,19,20]. To date, most natural enemy surveys have been oriented to larval parasitoids, and few were focused on other insect states such as egg, pupa, or adult [18,19,21]. In the same way, little attention has been given to generalist predators of the pest. In Mexico, 88 natural enemies’ species have been reported, including in six insect orders, two of Arachnida, and two species of birds [20]. Among the most common predators are several species of *Zelus*, *Orius*, *Podisus*, and *Chrysoperla* [21,22]. Some studies have also been done with earwigs like *Doru* sp. (Dermaptera: Forficulidae), which have a high predatory potential of eggs and larvae of early instars of this pest [23].

So far, there are records of seven parasitoid species of fall armyworm eggs in Mexico [20] of 10 species that have been registered worldwide [24]. Among these species *Trichogramma pretiosum* Riley (Hymenoptera: Trichogrammatidae) and *Telenomus remus* Nixon (Hymenoptera: Platygastridae) are the most used in biological control of *S. frugiperda* in Latin American countries such as Venezuela, Colombia, and Brazil [25]. Augmentative biological control programs against *S. frugiperda* in Mexico are mainly conducted using the egg parasitoid *T. pretiosum*, which has been produced and released since 1963 [20]. Although there are partially satisfactory results with the massive release of this parasitoid [26], evaluations are needed to assess if this *Trichogramma* species substantially reduces the pest in the field, because apparently its use is based on the ability to produce the species at low cost and in large quantities [27]. It is very likely that natural control of the pest will occur due to the decrease in the use of pesticides and the action of the native beneficial fauna in the area where it has been released, and not because of *T. pretiosum* release. [28]

Considering that native parasitoids of *S. frugiperda* eggs are still not well studied in Mexico, and that there is still no efficient egg parasitoid for the control of this pest, this work had two objectives. The first was to look for these parasitoids, using the sentinel technique in maize and sorghum crops, and to assess the natural control by predators in Central Mexico. The second was to compare the egg parasitism rate of two *Trichogramma* species on *S. frugiperda* in laboratory and field cages.

## 2. Materials and Methods

### 2.1. Insects

We established a laboratory colony of *S. frugiperda* with 100 larvae provided by the area of Insect Toxicology of the Colegio de Postgraduados in Texcoco, Estado de Mexico, which had been maintained for more than 15 generations on an artificial diet. The larvae were allowed to develop on an artificial diet following methodology developed by Poitout and Bues [29] with slight modifications (A. Pérez-Panduro, Colegio de Postgraduados, personal communication). The larvae were managed individually in plastic containers with approximately 10 g of artificial diet.

To obtain *S. frugiperda* eggs, groups of 30–40 adults were placed in a brown paper bag of 1.5 L capacity, which acted as a cage and as an oviposition substrate. The bag was changed daily to obtain egg masses less than 24 h old, the egg masses were attached to the paper bag and those pieces of paper were cut to facilitate their handling. To feed the adults, each paper bag had a plastic container with cotton saturated in sugar solution (10%). The fall armyworm adults in each bag were changed daily. The entire breeding process was developed at 25 ± 2 °C, 75% ± 5% RH, and 12:12 h Light:Dark (L:D). A portion of the eggs was used to maintain the colony in the laboratory, the remainder for field and laboratory experiments. 

### 2.2. Sentinel Technique to Detect Egg Parasitoids in Cereal Fields

The sentinel technique (mass exposure of eggs in the field) was used for the recovery of parasitoids in the field following Da Silva methodology [30]. Egg masses, that were used as sentinels, were kept refrigerated, at 8 °C, until they were used in the field. From May to October of 2017 and 2018, twelve locations were chosen to conduct field surveys for egg parasitoids and predators. Sentinels were placed in each location for 3 to 4 consecutive weeks during the susceptible stage of the culture. These included plots of maize and sorghum in different agro-ecological areas of Guanajuato State, Mexico. In each locality, a plot with an average surface area between 1 and 2 ha was chosen, the agronomic management of each one was recorded, and an average of 62 sentinels per location were placed. The pieces of paper with the sentinels were placed individually on the underside of the second leaf near the base of a maize or sorghum plant. Half of the sentinels were stapled on the underside of the leaves; the rest were placed, inside a 30 mL plastic cup, one sentinel per cup, and hung with a 10 cm tread from the plant in order to prevent predation.

The sentinels were left 48 h in the field, after this time they were checked one by one to look for signs of predation and the level of predation was recorded. If natural enemies were found on them, they were collected in 70% ethanol for further identification. Egg masses were removed and placed individually in ventilated 25 mL plastic vials, and kept at 25 ± 2 °C, 75 ± 5% RH, and 12:12 h L:D to wait for the parasitoids or the host larvae emergence.

### 2.3. Parasitoid Identification

Adult parasitoids, obtained from sentinels, were prepared for identification following the technique of Noyes [24], with some modifications suggested by Myartseva (Universidad Autónoma de Tamaulipas, personal communication). The keys of Trichogrammatidae from Pinto were used for species identification [31]. Voucher specimens of all natural enemies collected were deposited in the Colegio de Postgraduados Insect Collection.

### 2.4. Parasitism of Two Trichogramma Species, Laboratory Test

Parasitism of *T. pretiosum* was compared with the parasitoid recovered from the field (*Trichogramma atopovirilia* Oatman and Platner) under laboratory conditions. The first species was obtained from a commercial supplier (Organismos Benéficos para la Agricultura S.A. de C.V.—OBASA), and the second from the colony established in laboratory with material collected from the field, which was reared on eggs of *S. frugiperda*. In the parasitism experiments, females less than 24 h old and without experience in oviposition were used; the sex of the insects was assessed using a stereomicroscope (male antenna with relatively long setae).

The parasitism percentage of each *Trichogramma* species was assessed using a 50 egg mass of *S. frugiperda*, using a Petri dish (3 cm in diameter) as an arena. One female and a male parasitoid were released in each arena. The eggs were exposed for 24 h to the parasitoids, and 40 repetitions were performed per treatment (species). After this period the egg masses were kept in the environmental conditions already described. After six days, the parasitism was assessed by each treatment was determined. To evaluate parasitism, images of each egg mass were first obtained, using a scanner, and subsequently each image was analyzed with the ImageJ program using, in Edit-Options, the “point” tool [32]. The percentage of parasitism for each *Tricogramma* species in each egg mass was assessed by counting the total number of *S. frugiperda* eggs in the mass and the parasitized eggs, and this percentage was transformed by the arcsine function to meet the normality assumptions, and a Student’s t-test was performed with the R version 3.6 program [33].

### 2.5. Parasitism in Field Cages

The experiment consisted of two species of parasitoids and three release doses (ratio of one, two, and three unsexed adults of *Trichogramma* per one host egg), including a control without parasitoid release. The experimental unit consisted of a wooden frame cage with mesh screen (3 × 3 × 1.8 m) where three maize rows were isolated, 10 plants in each row, in which five sentinels (50 *S. frugiperda* eggs each) and a total of 250 eggs per cage were placed. To ensure an exact amount of eggs, the sentinel masses were checked under a microscope and leaving only 50 eggs per mass. A factorial design (3 doses × 2 *Trichogramma* species) was used in a completely randomized experimental design. Each treatment had six repetitions and the whole experiment was performed in a field plot less than 1 ha. In the field, the cages were first established to isolate the groups of plants. Then using a motorized vacuum, unwanted organisms (pests and entomophagous) were removed from inside the cages, for doing that we vacuumed 10 min in each cage. Within each cage, five masses of *S. frugiperda* eggs were randomly placed on the underside of the leaves of five plants, following the technique of Da Silva [30]. The parasitoids, either *T. pretiosum* or *T. atopovirilia* depending on the treatment, were released immediately after. The sentinels were exposed for 48 h. After this period, the masses were removed and kept at 25 ± 2 °C and 75 ± 5% RH. Parasitism was assessed by counting the total number of *S. frugiperda* eggs and the parasitized eggs in the sentinels by capturing images with a scanner, and subsequently analyzing each image of a mass of eggs with the ImageJ program [32]. Data were analyzed by logistic regression at a significance level of α = 0.05. The comparison of means was performed using the contrast method. Statistical analyses were performed with the R version 3.6 program [33].

## 3. Results

### 3.1. Natural Enemies of Egg Masses of S. frugiperda

During May to October periods of 2017 and 2018, a total of 4323 *S. frugiperda* sentinels were exposed at 12 field sites in Guanajuato (Table 1). From these, *T. atopovirilia* was recovered from three sites and *Chelonus insularis* Cresson from two sites. The percentage of natural parasitism due to *T. atopovirilia* was 2.8% in maize crops in Comonfort in the 2017 cycle, 3.75% in maize in Acámbaro, and 2.47% in sorghum in Tarimoro during the 2018 cycle; all those places are located in Guanajuato.

It was common to observe, at the time of sentinel collection, predators on the *S. frugiperda* egg masses; the most frequently predators were: *Doru taeniatum* Dohrn, *Chrysoperla carnea* Stephens, *Hippodamia convergens* Guérin-Meneville, *Collops* sp., *Solenopsis* sp., *Balaustium* sp., *Podisus* sp., and *Polistes* sp. 

The predator complex caused different predation rates (Table 1, Figure 1). We also found that egg predation was different among locations; the lowest was registered in Villadiego, Valle de Santiago (7%) and the highest in Cortazar, Cortazar (63%). 

### 3.2. Parasitism of Two Trichogramma Species in Laboratory Conditions

The percentage of parasitism was different among Trichogramma species (t = 6.728, df = 78; *p* ≤ 0.001). *T. atopovirilia* had an average parasitism of 70.14% while *T. pretiosum* had an average parasitism of 29.23%; in several *T. pretiosum* trials no parasitism was recorded.

### 3.3. Parasitism in Field Cages

The percentage of parasitism in field cages was different between treatments (χ^2^ = 754.72; df = 36; *p* ≤ 0.001). Total parasitism did not exceed 8% in any of the two species, but *T. atopovirilia* parasitized a greater number of hosts in all release doses (Figure 2). Parasitism at the dose of two *T. atopovirilia* parasitoids per egg was slightly, but significantly, greater than parasitism at a dose of one parasitoid per egg, but did not differ from parasitism at the dose of three *T. atopovirilia* parasitoids per egg (Figure 2).

## 4. Discussion

From more than 4000 sentinels (egg masses of *S. frugiperda*) distributed in two agricultural cycles in Guanajuato, only *T. atopovirilia* was recovered from three sites. *T. atopovirilia* is a parasitoid that is spread from the South of USA to Brazil [34]. In Mexico, it had been reported parasitizing eggs of the sugarcane borer *Diatraea* sp. throughout eleven states on the coasts [35,36], and only in *S. frugiperda* eggs in Veracruz [37]. Now its presence is confirmed for the first time in Central Mexico associated with *S. frugiperda* in maize and sorghum fields.

Guanajuato is a state in Central Mexico which has an important role of producing maize and sorghum and exporting vegetables. Additionally, there is a record of constant releases of *T. pretiosum* to combat *S. frugiperda* in maize or even some vegetables [38]. Despite these practices, *T. pretiosum* was not recovered from maize or sorghum fields in Guanajuato using sentinels’ eggs. This could have at least three explanations. First, the releases of *T. pretiosum* in the study region were not being detectably established in the field. Second, although no insecticides were used in the 12 search sites (except for one site), it is not ruled out that the use of insecticides in the region has influenced the poor recovery of more parasitoids. It is well known that the application of insecticides can have lethal and sublethal effects in a crop mosaic [39], as was characteristic in the study region. The third possible reason may be related to sampling. Although it is true that *S. frugiperda* can be on the crop edges or in the weeds throughout the whole year, the abundance of this pest and its eggs occurs mainly in the first two months of the establishment of the crop; therefore, in future works, it is recommended to concentrate the greatest collection effort in March and April, and August and September. Coincidentally, the recovery of *T. atopovirilia* in the field happened on October 2017, and September and October 2018.

Generalist predators were the most abundant natural enemies in maize and sorghum fields in Guanajuato. Egg mass predation was often related to the degree of crop management; for example, in Cortazar, a site with traditional management and presence of weeds, total predation levels exceeded 21%, and partial predation levels above 41%, this means almost 63% of masses had some level of predation. In this locality, the most frequent predators were mites of the genus *Balaustium* (19 events), followed by adults of *H. convergens* and the earwig (*D. taeniatum*). On the contrary, in Roque, a place with conventional management, without the presence of weeds, only 13% of total and partial predation was reported. It is likely that the presence of weeds, on plots where “weed control” was not carried out, has led to their abundance, since weeds serve as shelter and/or food sources providing nectar, pollen, or alternate prey [40]. Similarly, Hoballah [37] and Andrews [21] reported *S. frugiperda* egg predators in maize-bean associations around plots with weeds; including nymphs and adults of *Doru* sp., lacewings and pirate bugs *Orius* sp. *Doru lineare* can consume up to 438 eggs of *S. frugiperda* during its life, and also consume neonatal larvae of this pest [41]; however, it can feed on maize pollen and some farmers often consider it as a pest [21].

In field trips to Guanajuato, a mite of the genus *Balaustium* predating eggs of *S. frugiperda* was very common. *B. putmani* Smiley was reported predating eggs and larvae of first instar of *S. frugiperda* in apple orchards [42]; however, this is the first record of *Balaustium* sp. feeding on *S. frugiperda* eggs in maize. Due to its frequency found in the field, it is advisable to continue assessing its abundance and importance as natural control of *S. frugiperda*, as well as the role of the predator complex of the pest. 

Laboratory parasitism of the two *Trichogramma* species was markedly different; *T. atopovirilia* showed better performance than *T. pretiosum*. The parasitism percentages of *T. pretiosum* were low (29.23%) and there were several events where no parasitism was recorded. The quality of this material probably had an adverse effect to perform parasitism, although the influence of the scales on the eggs of the host is not ruled out. This explanation is related to the results of Beserra and Parra [43], whose findings indicated that *T. atopovirilia* had a greater capacity for parasitism than *T. pretiosum* on overlapping eggs with and without *S. frugiperda* scales.

Although high doses of parasitoids were released in field cages, high levels of parasitism were not recorded. This could be due to typical rains during the releases in the cages, which could directly affect the flight and survival of the parasitoids. The use of sentinels could be another factor that affected the performance of parasitoids, since stimuli such as the sexual pheromone of *S. frugiperda*, or the volatile of the plant induced by oviposition, were probably not present. Parasitoids of the genus *Trichogramma* use these infochemicals as a way to locate their hosts [44,45].

Biological control of *S. frugiperda* has been attempted in Mexico, mainly with mass release of *T. pretiosum*, but it has variable results, ranging from 2% to 70% parasitism [46,47]. One of the main problems faced by these parasitoids is the barrier of scales that the host’s female deposits over the eggs [48], and it seems that *T. pretiosum* does not have the capacity to overcome that barrier; on the other hand, *T. atopovirilia* has the capacity to overcome this barrier due to a greater aggressiveness and specificity to the host [43].

In Brazil, up to 63.2% of parasitism by *T. atopovirilia* was reported on egg masses of *S. frugiperda* of a single layer of eggs [43,49]. In this country, *T. atopovirilia* against *S. frugiperda*, and *T. pretiosum* against *Tuta absoluta* Meyrick, are released in more than 200,000 ha, which may foster interest in continuing to make assessments in other parts of the world [50]; but it should also encourage the search and evaluation of more natural enemies of *S. frugiperda* in the probable center of origin of the pest and crop, to contribute to the management of this key pest in about twenty countries on at least three continents. 

Without a doubt, it is necessary to have a clear, continuous, and permanent line of research in biological control of *S. frugiperda*, in order to provide elements to achieve better management. We must continue evaluations such as: (a) Biological conservation control, (b) the evaluation of the species of *T. pretiosum* and *T. atopovirilia* or any other should be considered to decide which is the most useful in an augmentative biological control program, (c) agricultural practices that contribute to improving biological control by conservation and making production systems more sustainable at any technological level.

## 5. Conclusions

Natural egg parasitism on *S. frugiperda* in Guanajuato was very low. *T. atopovirilia* was the only recovered species reaching 2.8%, and 3.75% of parasitism in maize in Comonfort and Acámbaro, and 2.47% in sorghum in Tarimoro.

The main mortality factor on egg masses was predation, and predators such as mites of the genus *Balaustium*, adults of *H. convergens,* and the earwig *D. taeniatum* were the most important. Together they achieved up to 63% predation in Cortazar. 

This is the first report of mites of the genus *Balaustium* predating eggs of *S. frugiperda*.

Laboratory and field experiments showed that *T. atopovirilia* had better performance than *T. pretiosum* in parasitizing *S. frugiperda* mass eggs. 

## Figures and Tables

**Figure 1 insects-11-00157-f001:**
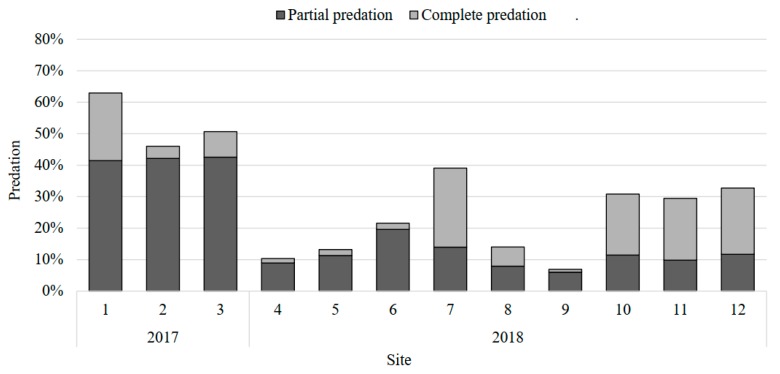
Predation level on *Spodoptera frugiperda* eggs in 12 Guanajuato localities, Central Mexico, during 2017 to 2018.

**Figure 2 insects-11-00157-f002:**
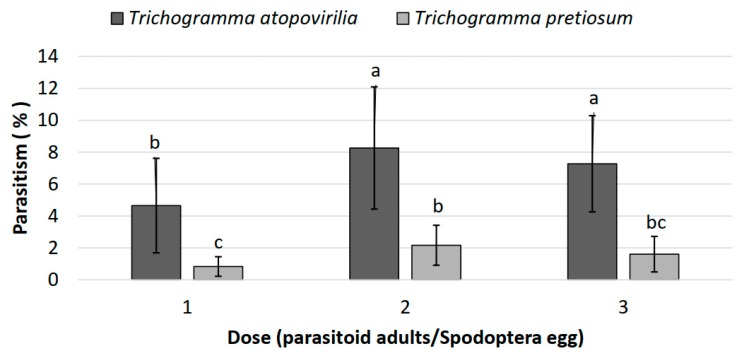
Parasitism in field cages by *T. atopovirilia* and *T. pretiosum* on *Spodoptera frugiperda* eggs. Same letter above the columns indicates non-significant differences between treatments.

**Table 1 insects-11-00157-t001:** Locality, municipality and quantity of recovered masses of *Spodoptera frugiperda* in Guanajuato, Mexico (spring–summer 2017–2018 cycle).

Site	Locality		Level of Predation in Sentinels		
		Year	Without Damage	Fully Consumed	Partially Consumed
1	Cortazar, Cortazar	2017	221	128	247
2	INIFAP, Campo Experimental Bajío, Celaya	2017	542	39	423
3	Morales, Comonfort	2017	436	72	376
4	Instituto Tecnológico de Roque, Celaya	2018	191	3	19
5	Roque, Celaya	2018	138	3	18
6	Neutla, Comonfort	2018	80	2	20
7	Empalme Escobedo, Comonfort	2018	131	30	54
8	Jaral del progreso, Jaral del progreso	2018	141	13	10
9	Villadiego, Valle de Santiago	2018	108	7	1
10	El Piloncillo, Acámbaro	2018	211	59	35
11	La Concepción, Acámbaro	2018	194	54	27
12	San Nicolás de la Condesa, Tarimoro	2018	195	61	34
Total			2588	471	1264

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
