# Peer review of "Egg Parasitoids Survey of Spodoptera frugiperda (Smith) (Lepidoptera: Noctuidae) in Maize and Sorghum in Central Mexico"

_insects, 2020, doi:10.3390/insects11030157_

Round 1
Reviewer 1 Report
This manuscript describes an investigation of potential native biological control agents for S. frugiperda in Mexico and compares parasitism rate in the lab and in field cages between T. pretiosum and T. atopovirilia (both egg parasitoids). The study will provide a useful contribution to the field. I have only minor comments.
1) Abstract: lines 25-26. State the actual results here. Stating only that the two treatments were different isn't very informative.
2) Abstract: lines 26-27. This final sentence should be revised to state how this study informs where researchers should go next. As written, the sentence is not particularly meaningful.
3) Introduction: line 72. Revise as "Considering that native parasitoids of S. frugiperda eggs are still not well studied in Mexico, ..."
4) Introduction: line 76. Be more specific about what you are comparing - egg parasitism rate?
5) Materials and Methods: line 79. Revise as "We established a laboratory colony of S. frugiperda with 100 larvae ..."
6) Materials and Methods: line 81. Instead of "reproduced on", revise as "allowed to develop on"
7) Materials and Methods: lines 103-104. The plastic cup methodology is unclear. Were these cups somehow attached to the undersides of the leaves?
8) Materials and Methods: lines 133-134. I do not quite understand how you determined the number of parasitoids to release. Did you estimate the # of eggs in each egg mass and then released parasitoids at the rate of 1x, 2x, and 3x the estimated # of eggs?
9) Results: line 170. Provide the degrees of freedom for this t-test.
10) Results: line 174. Provide the statistics (other than a p-value) for the logistic regression.
11) Results: lines 176-178. What you've stated here does not match what is shown in Figure 2. Firstly, according to the letters above the columns in Figure 2, T. atopovirilia parasitized a greater # of eggs at all 3 doses of parasitoids (not just two as stated in the text). Secondly, the final sentence here should be revised as "Parasitism at the dose of two T. atopovirilia parasitoids per egg was slightly, but significantly, greater than parasitism at a dose of one parasitoid per egg, but did not differ from parasitism at the dose of three T. atopovirilia parasitoids per egg. Further, parasitism at a dose of three parasitoids per egg was no greater than parasitism at a dose of one parasitioid per egg."
12) Figure 2 legend: line 182. State what the letters above the columns indicate.
13) Discussion: line 194. Italicize T. pretiosum.
14) Discussion: line 201. Insert a period after 'sampling' and begin a new sentence with 'Although'.
15) Discussion: line 210. 'prey' instead of 'preys'
16) Discussion: line 214. 'consume' instead of 'prey'
17) Discussion: lines 224-226. Unclear - provide more explanation.
18) Discussion: line 230. Did it rain more than usual?
19) Conclusions: lines 249-255. Run-on sentence. Break into two or three sentences.
Author Response
1) Abstract: lines 25-26. State the actual results here. Stating only that the two treatments were different isn't very informative. Response 1: Done. See lines 25-28
2) Abstract: lines 26-27. This final sentence should be revised to state how this study informs where researchers should go next. As written, the sentence is not particularly meaningful. . Response 2: Done. See lines 25-28
3) Introduction: line 72. Revise as "Considering that native parasitoids of S. frugiperda eggs are still not well studied in Mexico, . . Response 3: Done. See line 74.."
4) Introduction: line 76. Be more specific about what you are comparing - egg parasitism rate? . Response 4: Done. See lines 77-78
5) Materials and Methods: line 79. Revise as "We established a laboratory colony of S. frugiperda with 100 larvae ..." . Response 5: Done. See lines 81
6) Materials and Methods: line 81. Instead of "reproduced on", revise as "allowed to develop on" . Response 6: Done. See lines 83
7) Materials and Methods: lines 103-104. The plastic cup methodology is unclear. Were these cups somehow attached to the undersides of the leaves? . Response 7: Done. See lines 106-107
8) Materials and Methods: lines 133-134. I do not quite understand how you determined the number of parasitoids to release. Did you estimate the # of eggs in each egg mass and then released parasitoids at the rate of 1x, 2x, and 3x the estimated # of eggs? Response 8: Done. See lines 142-144
9) Results: line 170. Provide the degrees of freedom for this t-test. Response 9: Done. See line 177
10) Results: line 174. Provide the statistics (other than a p-value) for the logistic regression. Response 10: Done. See lines 181-182
11) Results: lines 176-178. What you've stated here does not match what is shown in Figure 2. Firstly, according to the letters above the columns in Figure 2, T. atopovirilia parasitized a greater # of eggs at all 3 doses of parasitoids (not just two as stated in the text). Secondly, the final sentence here should be revised as "Parasitism at the dose of two T. atopovirilia parasitoids per egg was slightly, but significantly, greater than parasitism at a dose of one parasitoid per egg, but did not differ from parasitism at the dose of three T. atopovirilia parasitoids per egg. Further, parasitism at a dose of three parasitoids per egg was no greater than parasitism at a dose of one parasitioid per egg." Response 11: Done. See lines 183-186
12) Figure 2 legend: line 182. State what the letters above the columns indicate. Response 12: Done. See line 191
13) Discussion: line 194. Italicize T. pretiosum. Response 13: Done. See line 201
14) Discussion: line 201. Insert a period after 'sampling' and begin a new sentence with 'Although'. Response 14: Done. See line 208
15) Discussion: line 210. 'prey' instead of 'preys' Response 15: Done. See lines 223
16) Discussion: line 214. 'consume' instead of 'prey' Response 16: Done. See line 224
17) Discussion: lines 224-226. Unclear - provide more explanation. Response 17. Done. See lines 234-240
18) Discussion: line 230. Did it rain more than usual? Response 18: Done. See line 242
19) Conclusions: lines 249-255. Run-on sentence. Break into two or three sentences. Response 19: This paragraph was changed at the end of the discussion at the request of one of the revisers. See lines 260-266
Reviewer 2 Report
COMMENTS TO THE MS insects-643032
General comment
Jaraleño-Teniente et al. carried out field (maize/sorghum), semi-field (maize) and laboratory experiments to assess parasitoids and predators of Spodoptera frugipera eggs. In all experiments, authors used the egg sentinel technique to detect the arthropods that parasitize or predate the S. frugipera eggs. The experiments were very clear and followed standard procedures. However, I miss some novelty, and results obtained in this study could be predicted from existing literature.
I found the Introduction well written and concise, M&M are ok, but some details could be improved. Results and discussion in my opinion need some revision and be improved.
Although, I found lack of novelty in this study, I recommend minor revisions to this Ms for publication in insects.
Specific comments
L 90 and 109: instead of L:O, change to L:D
L 96: spring-summer, specify dates, months
L101-104: specify growth stage
L 117-118: T. atopovirilia is the first time that is mentioned, please provided full genus and author
L 137: 3x2 or 4x2
L 138: parasitism in cages was done only in 1 field, please clarify it
L 139: I have some doubts that all emtomophagous could be removed from the cages
L141-142: Both parasitoids were released in the same cages, was not clear for me!
Results
L150: summer-autumn? In M&M and table 1 is reported spring-summer?
L150: change Spodoptera per S.
L151: T. atopovirilia
L158-159: Discussion, maybe!
L160- 166; Management could be discussed, but not here, please in this section report only the results. In M&M no information about crop management on each site is given; and also this variable was not taking into account in the data analysis.
In my opinion, the results are a bit confused and not so well written as introduction. Figures need to be uniformed, and resolution improved,
Discussion
Discussion is too long for the results obtained.
L187: T. atopovirilia
L 193: T. pretiosum should be in italic
L208-210: this paper maybe could be useful here: Albajes et al 2009. Responsiveness of arthropod herbivores and their natural enemies to modified weed management in corn.
L222-224: T. atopovirilia was reared in S. frugiperda eggs could it affect results, if I am not wrong there are some literature about this topic.
L224-226: I did not get the idea here!
L 243: Tuta absoluta (author)
Conclusions
None of the conclusions presented come from this study! Please rewrite it, and conclude based on your results!
Author Response
L 90 and 109: instead of L:O, change to L:D. Response 1: Done. See lines 92 and 112
L 96: spring-summer, specify dates, months. Response 2: Done. See line 98
L101-104: specify growth stage Response 3: I consider it unnecessary since it was on different collection dates
L 117-118: T. atopovirilia is the first time that is mentioned, please provided full genus and author Response 4: Done. See line 121
L 137: 3x2 or 4x2. Response 5: Done. See lines 142-143
L 138: parasitism in cages was done only in 1 field, please clarify it. Response 6: Done. See line 146-147
L 139: I have some doubts that all emtomophagous could be removed from the cages. Response 7: Done. See line 149
L141-142: Both parasitoids were released in the same cages, was not clear for me! Response 8: Done. See line 151
Results
L150: summer-autumn? In M&M and table 1 is reported spring-summer? Response 9: Done. See line 161
L150: change Spodoptera per S. Response 10: Done. See line 161
L151: T. atopovirilia Response 11: Done. See line 162
L158-159: Discussion, maybe! Response 12: Done. We move this paragraph to discussion See lines 215-221
L160- 166; Management could be discussed, but not here, please in this section report only the results. In M&M no information about crop management on each site is given; and also this variable was not taking into account in the data analysis. Response 13: Done. We move this paragraph to discussion See lines 215-221
In my opinion, the results are a bit confused and not so well written as introduction. Response 14: Done. We improve this section Figures need to be uniformed, and resolution improved, Figures will be send at best quality if the manuscript is accepted
Discussion
Discussion is too long for the results obtained. Response 15: In our opinion it is not too long
L187: T. atopovirilia Response 16: Done. See line 194
L 193: T. pretiosum should be in italic Response 17: Done. See line 201
L208-210: this paper maybe could be useful here: Albajes et al 2009. Responsiveness of arthropod herbivores and their natural enemies to modified weed management in corn. 40. Response 18: We think that Norris, R.F.; Kogan´s review is more inclusive and addresses the topic of the recommended document
L222-224: T. atopovirilia was reared in S. frugiperda eggs could it affect results, if I am not wrong there are some literature about this topic. Response 19: it could be an element of discussion but we consider it is unnecessary to open another section so as not to make the document longer
L224-226: I did not get the idea here! Response 20: Done see lines 234-240
L 243: Tuta absoluta (author) Response 21: see line 255
Conclusions
None of the conclusions presented come from this study! Please rewrite it, and conclude based on your results! Response 22: see line 269-277
Reviewer 3 Report
Title: Egg parasitoids survey of spodoptera frugiperda (Smith) (Lepidoptera: Noctuidae) in Maize and Sorghum in Central Mexico
The manuscript attempted to survey parasitoids and predators of fall armyworm in grains. The paper has good information and certainly should be published. The statistical analysis seems to be appropriate. Some questions, comments and suggestions below.
Line 13: add family and order name after species name
Line 31: why the parenthesis is not present for “Smith”
Line 41: Organ-synthetic insecticides? Please make this clear with examples.
Line 94: How many times sentinel egg masses were deployed in the 12 sites?
Lines 103-104: Could please explain a little bit on “…placed in 30-ml plastic cups (No.0) to prevent predation.” No. 0 cup, not clear. How this would prevent predation?
Line 122: Citation required for sex determination. Presence or ovipositor? Please clarify.
Line 123: How many eggs were there in an egg mass? Please present mean number of eggs in the egg mass.
Line 125: replace “hours with “h”.
Line 129: Explain how the “ImageJ” software work?
Lines 133-134: Please say what was the sex of the parasitoid used in one, two and three treatments.
Line 137: Please say it again what were the factors and levels for “3x2”. This will bring clarity.
Lines 156-158: How the authors determined the species of predators visiting the egg masses? The predators should have left the scene when authors went to collect the egg cards, right? Please explain.
Table 1: Please check of “predated” is a right word accepted in English dictionary? If not, replace it with “predation”.
Line 188: Replace “southern” with “South”
Line 189: Please say what Diatrea sp. is? Insert family and order.
Author Response
Line 13: add family and order name after species name Response 1: this is not common in the abstracts of the articles published in this journal
Line 31: why the parenthesis is not present for “Smith”. Response 2: see line 32
Line 41: Organ-synthetic insecticides? Please make this clear with examples. Response 3: see line 42
Line 94: How many times sentinel egg masses were deployed in the 12 sites? Response 4: Done see lines 106-107
Lines 103-104: Could please explain a little bit on “…placed in 30-ml plastic cups (No.0) to prevent predation.” No. 0 cup, not clear. How this would prevent predation? Response 5: Done see lines 106-107
Line 122: Citation required for sex determination. Presence or ovipositor? Please clarify. Response 6: Done see lines 126
Line 123: How many eggs were there in an egg mass? Please present mean number of eggs in the egg mass. Response 7; as it was indicated in the text, masses of 50 eggs were used see line 127
Line 125: replace “hours with “h”. Response 8: Done see line 129
Line 129: Explain how the “ImageJ” software work? Response 9: Done see line 133-134
Lines 133-134: Please say what was the sex of the parasitoid used in one, two and three treatments. Response 10: Done see line 140
Line 137: Please say it again what were the factors and levels for “3x2”. This will bring clarity. Response 11: Done see line 145
Lines 156-158: How the authors determined the species of predators visiting the egg masses? The predators should have left the scene when authors went to collect the egg cards, right? Please explain. Response 12: Done see line 166
Table 1: Please check of “predated” is a right word accepted in English dictionary? If not, replace it with “predation”. Response 13: We change predated for consumed see line 175
Line 188: Replace “southern” with “South”
Line 189: Please say what Diatrea sp. is? Insert family and order. Response 13: Done see line 196